

# Discounted Mean-Field Game model of a dense static crowd with variable information crossed by an intruder

Matteo Butano[1,2⋆], Cécile Appert-Rolland[2] and Denis Ullmo[1]

**1** Université Paris-Saclay, CNRS, LPTMS, 91405, Orsay, France
**2** Université Paris-Saclay, CNRS, IJCLab, 91405, Orsay, France

⋆ matteo.butano@universite-paris-saclay.fr

## Abstract

It was demonstrated in [1] that the anticipation pattern displayed by a dense crowd crossed by an intruder can be successfully described by a minimal Mean-Field Games model. However, experiments show that changes in the pedestrian knowledge significantly modify the dynamics of the crowd. Here, we show that the addition of a single parameter, the discount factor $\gamma$, which gives a lower weight to events distant in time, is sufficient to observe the whole variety of behaviors observed in the experiments. We present a comparison between the discounted MFG and the experimental data, also providing new analytic results and insight about how the introduction of $\gamma$ modifies the model.



# 1   Introduction

As discussed by Hoogendorn et al. [2], pedestrian motion is a multi-scale endeavour that needs to be described at different levels, and one typically distinguishes the strategic level (what are the goals of the travel and the general timing), the tactical level (which route to take) and the operational level (how to move on that route). If the strategic and tactical levels naturally assume some form of long term optimisation, it is on the other hand quite generally expected that the operational level, especially for dense crowds, could be described by dynamical models involving physical and social forces [3], and possibly a short term (up to the next collision) anticipation [4–6]. However, the comparison of experimental data from a control experiment involving a static dense crowd [7] with various simulation models showed that [1], in such cases, it appears necessary to include long term anticipation in order to reproduce the observed experimental features. This could be achieved with a very simple Mean-Field Game model.

Mean-Field Games, introduced by Lasry and Lions [8, 9] as well as Huang et al. [10] is a mathematical formalism in which both anticipation and competitive optimization between the agents, here pedestrians, are accounted for at the mean field level. The fact that it may provide a good description of human behaviour is a priori non trivial for at least two reasons. The first one is that the agents clearly cannot compute the predictions of the Mean-Field Games model, the second is that they usually do not have the perfect information assumed in the model.

The answer to this first concern is, as in many game theoretical context, that we expect the behaviour of the agents to be learnt rather than computed. In the same way that we do not need to use the laws of dynamics to know where a ball thrown in the air will land, there are in daily life many circumstances which are familiar enough that we can, by instinct, predict the behavior of the crowd and act accordingly. It is clearly in these circumstances that we can expect a Mean-Field-Game description to apply, and this was the case in the experimental context of [7]. The second of these concerns, which will be the main focus of this paper, was actually addressed experimentally in [7]. The experimental setup studied there consisted in the crossing of a dense static crowd by a cylindrical intruder. However, the experiments have been performed in three configurations: with pedestrian facing the obstacle, being randomly oriented, or giving their back to it. Moreover, in the latter case pedestrians were asked not to anticipate, while no such instruction was given in the first two cases. Clearly, if, in this simple context, we can consider that in the first configuration, participants have perfect information about the incoming cylinder, this is not the case for the other two, and indeed, as seen on Fig. 1, the observed density maps, obtained by an average over repeated realizations of the experiment, clearly differ significantly in the three cases.

In [1], we focused only on the case where pedestrians were facing the obstacle (for an average density of $2.5 ped/m^2$, slightly lower than the one shown in Fig. 1(a), but displaying the same features). Observing Figure 1(a), we see that in the frontal configuration, individuals move in advance, temporarily accepting a higher local density in order to get rid of the obstacle sooner, showing evidence of a trade-off dynamics. The crowd's density increases on the sides and depletes in front of and behind the cylinder, with pedestrians moving outward

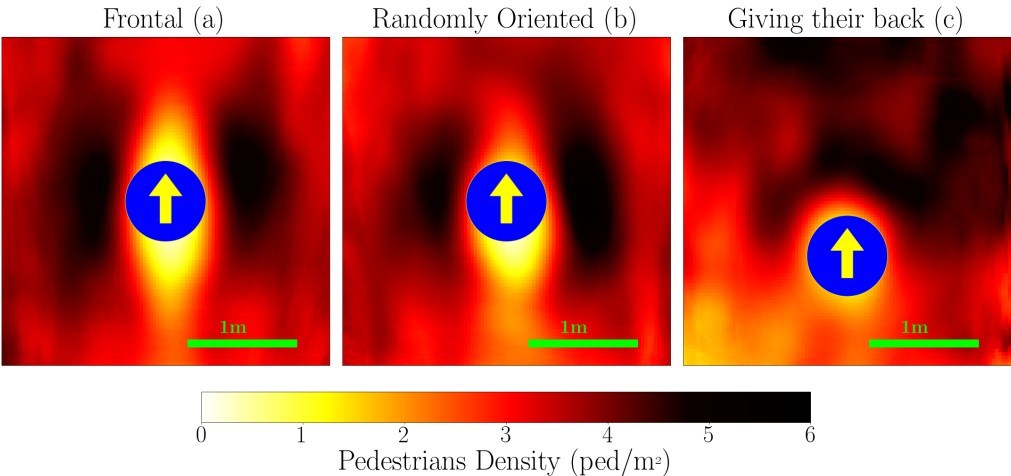

Figure 1: Experimental results of the passage of a cylindrical intruder (blue disc) through a static dense crowd for an average density of $\sim 3.5\, ped/m^2$ for three different configurations. In (a) pedestrians were all facing the obstacle, in (b) they were randomly oriented and in (c) participants were asked to give their back to it and not to anticipate. Data from [7].

to avoid its arrival and inward to regain a less congested position, always perpendicularly to the obstacle's motion. This clearly is an anticipatory dynamics. Indeed even pedestrians far from the cylinder know that it will arrive and, depending on its speed, they will start moving accordingly. However, when individuals are placed randomly, as in the second panel of Figure 1, we notice a shorter corridor of anticipation in front of the obstacles, but with similar accumulation patterns on its sides. When pedestrians in the experiment were asked not to anticipate and to give their back to the incoming intruder, effectively looking away from it, as the third panels of Figure 1 shows, we observe that the accumulation of the individuals moves from the sides to the front of the obstacle, similarly to what happens in analog configurations with granular matter [1].

The main goal of this paper is to demonstrate that at least in the experimental setup corresponding to Fig.1, this lack of information can be taken into account simply by introducing a *discount factor*, i.e. a term discounting the cost of actions in a future further than a certain cutoff, to the model already used in [1] to simulate the frontal case. Such discount factors are, for example, usually introduced in optimization problem spanning a long time period (typically in economic context) to express the unwillingness of agents to wait too long for a reward. Here however we use it as a way to simulate the shortsightedness of pedestrians. We shall in section 2 introduce the corresponding Mean-Field-Game model and give some discussion about its generic behavior. Then, section 4 will feature the comparison with the three experimental cases discussed above. We will first come back on the case where pedestrians were facing the obstacle, providing more details about the results presented in [1]. Then, the other experimental setups will be considered, and we will show how the introduction of the discount factor helps in modeling these configurations.

## 2 The mean-field games model

Mean-Field Games (MFG) constitutes a relatively new field of research. Its foundations are in the works of J.-M. Lasry and P.-L. Lions [8, 9], and of M. Huang, R. P. Malhamé and P. E. Caines [10]. During the years, many works have been focused on the mathematical properties

of MFG [11–15]. Although there are applications of MFG to pedestrian dynamics [16], to the best of our knowledge comparisons of crowds simulated with MFG to experimental data [1] are rare. A general and mathematically rigorous discussion of the foundations of MFG being found in the book [14], and a physicist-friendly version being exposed in [17], here we will limit ourselves to essentials.

## 2.1 The discounted equations

In the specific settings of our MFG model, each agent's *state variable* $\vec{X}(t) \in \mathbb{R}^2$, representing their position, evolves following the *Langevin equation*

$$\dot{\vec{X}} = \vec{a}(t) + \sigma\vec{\xi}(t), \tag{1}$$

where $\vec{\xi}(t)$ is a two dimensional Gaussian white noise of variance one, and $\vec{a}$ is the drift velocity, the *control parameter* that represents the strategy players choose by minimizing the discounted cost functional defined, in this case, as

$$c[\vec{a}](\vec{x},t) = \mathbb{E}\left\{\int_t^T \mathcal{L}(\vec{x},\tau)[m]e^{\gamma(t-\tau)}d\tau + e^{\gamma(t-T)}c_T(\vec{x}_T)\right\}, \tag{2}$$

where $c_T$ is a *terminal cost*, that could be used to represent a target for pedestrians, as for instance an exit, but which here is simply taken as $c_T = 0$. In this equation, $\gamma$ is the discount factor, and its inverse $1/\gamma$ defines an anticipatory time scale determining how far into the future agents will look while optimizing. Finally

$$\mathcal{L}(\vec{x},\tau)[m] = \frac{\mu}{2}(\vec{a}(\tau))^2 - V[m](\vec{x},\tau), \tag{3}$$

can be seen as the term describing the agents' preferences. In fact, the squared velocity tells that going too fast is detrimental, and that the best would be to stand still, but the presence of the external world, represented by the potential term

$$V[m](\vec{x},t) = gm(\vec{x},t) + U_0(\vec{x},t), \tag{4}$$

describing the interaction with the others and with the environment, cause agents to actually move. The main assumptions of MFG are that all agents are equal and the interaction with others is of mean-field type, determined only through the average density

$$m(\vec{x},t) = \mathbb{E}\left[\frac{1}{N}\sum_{i=1}^N \delta(\vec{x}-\vec{X}_i(t))\right]. $$

For $g < 0$, the first term in the right hand side of Eqs. (4), $gm(\vec{x},t)$, expresses the agents' desire to stay away from densely packed areas. On the other hand, the term $U_0(\vec{x},t)$ describes the interaction with the environment, and in the case of the crossing cylinder reads

$$U_0(\vec{x},t) = \begin{cases} +\infty, & \|\vec{x}-\vec{s}t\| < R, \\ 0, & \text{otherwise}, \end{cases} \tag{5}$$

where $\vec{s}$ is the velocity and $R$ the radius of the cylinder. A quantity of interest of the MFG is then the *value function*

$$u(\vec{x},t) = \inf_{\vec{a}} c[\vec{a}](\vec{x},t), \tag{6}$$

that, as shown in appendix A, can be obtained solving the *Hamilton-Jacobi-Bellman* equation

$$\begin{cases} \partial_t u = -\frac{\sigma^2}{2}\Delta u + \frac{1}{2\mu}(\vec{\nabla}u)^2 + \gamma u + V[m], \\ u(\vec{x}, t = T) = c_T(\vec{x}). \end{cases} \tag{HJB}$$

This is a backward equation, that is solved starting from the terminal condition expressed by the terminal cost $c_T(\vec{x})$. The optimal choice of the control parameter is $\vec{a}^* = -\vec{\nabla}u/\mu$. Given the stochastic evolution of each player's state variable, the corresponding average density evolves following the Kolmogorov-Fokker-Plank equation

$$\begin{cases} \partial_t m = \frac{\sigma^2}{2}\Delta m + \frac{1}{\mu}\vec{\nabla}\cdot(m\vec{\nabla}u), \\ m(\vec{x}, t = 0) = m_0(\vec{x}), \end{cases} \tag{KFP}$$

a forward equation solved starting from an initial density profile.

As discussed extensively in [1, 17], in the $\gamma = 0$ limit the sytem of equations (HJB)-(KFP) can be cast in a form analog to the non-linear Schrödinger equation through the combination of a Cole-Hopf transformation and of an hermitization of the KFP equation. We show in appendix B how this formulation evolves for a general $\gamma$.

## 2.2 The ergodic state

### 2.2.1 Without discount factor

When $\gamma = 0$, i.e. when the anticipation time $1/\gamma$ is not bounded, we know from [18], that under certain hypothesis, among which a large optimisation time $T$ and the time independence of the potential $V$, and for times far from the beginning and the end, an ergodic or stationary state exists. Such state is characterised by the fact that its observable quantities, i.e. the optimal velocity field and the density field, do not depend on time anymore. In particular, for the velocity this means that

$$\partial_t \vec{a}_e^*(\vec{x}, t) = -\frac{1}{\mu}\partial_t\vec{\nabla}u^e(\vec{x}, t) = 0 \qquad \implies \qquad u^e(\vec{x}, t) = u^e(\vec{x}) + u^e(t), \tag{7}$$

where the superscript $e$ refers to the ergodic state. Moreover, again from [18] we also know that for $\gamma = 0$, $\vec{u}(t)_e = -\lambda t$, where $\lambda$ is a parameter to be estimated. In the settings of our experiment, in the frontal case, so that we can use $\gamma = 0$, we can determine the value of $\lambda$ by looking at what happens far enough from the cylinder for the density to be unaffected by its presence. In fact, away from the obstacle we expect $m_e(\vec{x}, t) \simeq m_0$, $\forall t \in [0, T]$ and the optimal strategy is just to stay at rest, meaning $\vec{a}^* = 0$. Therefore, using definition (2) with $\gamma = 0$ we have

$$u(\vec{x}, t) = -\int_t^T (gm_0)d\tau = \underbrace{-gm_0T}_{u^e(\vec{x})} + \underbrace{gm_0}_{-\lambda}t, \tag{8}$$

and thus $\lambda = -gm_0$.

### 2.2.2 With discount factor

Here we make the rather natural hypothesis that, for large $T$, such a stationary state exists also when $\gamma \neq 0$, i.e. for a finite anticipation time $1/\gamma$. We therefore assume that for intermediate times, there exists a state of the discounted system whose observables are independent on

time. If that is the case, (7) should still hold, and we can as before determine the time dependent part of $u_e$ by considering what happens far from the cylinder where the density remains homogeneous and the optimal strategy is simply to stand still. In this case (2) becomes

$$u(\vec{x},t) = -\int_t^T (gm_0)e^{-\gamma(\tau-t)}d\tau = \frac{gm_0}{\gamma}\left(e^{-\gamma(T-t)}-1\right). \tag{9}$$

We observe that if we fix $T$ and let $\gamma \to 0$, the right hand side of Eq. (9) becomes $-gm_0 T + gm_0 t$, recovering Eq. (8) and the $\gamma = 0$ case. On the other hand, if we fix $\gamma$ and let $T \to \infty$, we have that $u^e(t) \equiv 0$, and since $u^e(t)$ does not depend on the position, this must be true everywhere, meaning that when $\gamma \neq 0$

$$u^e(\vec{x},t) = u^e(\vec{x}), \tag{10}$$

and we can write both (HJB) and (KFP) in their stationary form as

$$0 = \frac{\sigma^2}{2}\Delta u^e - \frac{1}{2\mu}(\vec{\nabla}u^e)^2 - \gamma u^e(x) - V[m^e], \tag{11}$$

$$0 = \frac{\sigma^2}{2}\Delta m^e + \frac{1}{\mu}\nabla\cdot(m^e\nabla u^e). \tag{12}$$

## 2.3 Passing to the moving frame

When we introduced the ergodic state in subsection 2.2 we stressed the necessity of the absence of any explicit time dependence in the cost functional (2) for Eqs. (11)-(12) to be possible, which conflicts with the fact that we simulate the presence of the intruder by using a cylindrical potential whose position evolves with time. We thus change the reference frame in which we describe the experimental setting by passing from the point of view of the laboratory to the point of view of the cylinder, and define

$$\hat{u}(\vec{x}-\vec{s}t,t) = u(\vec{x},t), \quad \hat{m}(\vec{x}-\vec{s}t,t) = m(\vec{x},t).$$

In the framework of the cylinder the potential becomes $\hat{V}[\hat{m}] = g\hat{m} + \hat{U}_0(\vec{x})$, not depending explicitly on time anymore, and with

$$\hat{U}_0(\vec{x}) = \begin{cases} +\infty, & x < R, \\ 0, & \text{otherwise}, \end{cases} \tag{13}$$

where $R$ is the radius of the cylinder. For all other quantities we observe that, in general,

$$\partial_t f(\vec{x},t) = \frac{\mathrm{d}}{\mathrm{d}t}f(\vec{x},t) = \frac{\mathrm{d}}{\mathrm{d}t}\hat{f}(\vec{x}-\vec{s}t,t) = \partial_t\hat{f} - \vec{s}\cdot\vec{\nabla}\hat{f}, \tag{14}$$

where the total derivative is taken in the lab's reference frame. Using (14) in (HJB) and (KFP) gives us the time dependent version of the MFG equation in $\hat{u}$ and $\hat{m}$ variables in the moving frame

$$\partial_t\hat{u} - \vec{s}\cdot\vec{\nabla}\hat{u} = -\frac{\sigma^2}{2}\Delta\hat{u} + \frac{1}{2\mu}(\vec{\nabla}\hat{u})^2 + \gamma\hat{u} + \hat{V}[\hat{m}], \tag{15}$$

$$\partial_t\hat{m} - \vec{s}\cdot\vec{\nabla}\hat{m} = \frac{\sigma^2}{2}\Delta\hat{m} + \frac{1}{\mu}\nabla\cdot(\hat{m}\nabla\hat{u}), \tag{16}$$

whereas the corresponding equations for the ergodic state are

$$0 = -\frac{\sigma^2}{2}\Delta\hat{u}^e + \frac{1}{2\mu}(\vec{\nabla}\hat{u}^e)^2 + \vec{s}\cdot\vec{\nabla}\hat{u}^e + \gamma\hat{u}^e(x) + \hat{V}[\hat{m}^e], \tag{17}$$

$$0 = \frac{\sigma^2}{2}\Delta\hat{m}^e + \frac{1}{\mu}\nabla\cdot(\hat{m}^e\nabla\hat{u}^e) + \vec{s}\cdot\vec{\nabla}\hat{m}. \tag{18}$$

## 3 Parameter space of the model

In [1], it was shown that when $\gamma = 0$, the agents' dynamics is entirely characterized by two parameters, a length scale

$$\xi = \sqrt{\left|\frac{\mu \sigma^4}{2g m_0}\right|},$$

that by analogy with the non-linear Schrödinger context we refer to as the *healing length*, and a velocity scale

$$c_s = \sqrt{\left|\frac{g m_0}{2\mu}\right|},$$

that, for the same reason, we refer to as the *sound velocity*. Their physical meaning can be understood by imagining that the pedestrian crowd is subjected to a local perturbation. The healing length would then be the distance up to which the density would deform, and the sound velocity the speed at which the density would return to its initial unperturbed state when the perturbation is removed. If we then include the parameters $R$ and $s$ characterizing the intruding cylinder, we see that when $\gamma = 0$, our MFG model is entirely characterized by two dimensionless quantities, $\tilde{R} \equiv R/\xi$ and $\tilde{s} = s/c_s$.

As shown in Appendix D, the inclusion of a non-zero discount factor $\gamma$, which has the dimension of the inverse of a time, implies that the MFG model is now characterized by three dimensionless quantities, $\tilde{R}$, $\tilde{s}$ and a third one that we can take as either $\tilde{\gamma}^{(1)} \equiv (\xi/c_s)\gamma$ or $\tilde{\gamma}^{(2)} \equiv (R/s)\gamma$ (Note $\tilde{\gamma}^{(2)} = (\tilde{R}/\tilde{s})\tilde{\gamma}^{(1)}$). The first option, $\tilde{\gamma}^{(1)}$, compares the time scale associated with anticipation to the one of the crowd dynamics, while $\tilde{\gamma}^{(2)}$ measures it in terms of the time scale characterizing the cylinder.

Fig. 2 shows the numerical solution of the stationary equations (17) and (18) (see appendix C for a brief description of the numerical implementation) for four choices of $(\tilde{R}, \tilde{s})$ and different values of $\gamma$. Note that in the $(\tilde{R} \gg 1, \tilde{s} \gg 1)$ and $(\tilde{R} \ll 1, \tilde{s} \ll 1)$ quadrants, we have assumed $\tilde{R} \sim \tilde{s}$, so that $\tilde{\gamma}^{(1)} \sim \tilde{\gamma}^{(2)}$, and both of them are therefore either large together or small together. However in the quadrant $(\tilde{R} \gg 1, \tilde{s} \ll 1)$, we have $\tilde{\gamma}^{(2)} = (\tilde{R}/\tilde{s})\tilde{\gamma}^{(1)} \gg \gamma^{(1)}$, so we have distinguished the three possible cases $(\tilde{\gamma}^{(1)} \ll 1, \tilde{\gamma}^{(2)} \ll 1)$, $(\tilde{\gamma}^{(1)} \ll 1, \tilde{\gamma}^{(2)} \gg 1)$, and $(\tilde{\gamma}^{(1)} \gg 1, \tilde{\gamma}^{(2)} \gg 1)$; and in the same way in the quadrant $(\tilde{R} \ll 1, \tilde{s} \gg 1)$ where $\tilde{\gamma}^{(2)} \ll \gamma^{(1)}$ we distinguish the three cases $(\tilde{\gamma}^{(1)} \ll 1, \tilde{\gamma}^{(2)} \ll 1)$, $(\tilde{\gamma}^{(1)} \gg 1, \tilde{\gamma}^{(2)} \ll 1)$, and $(\tilde{\gamma}^{(1)} \gg 1, \tilde{\gamma}^{(2)} \gg 1)$.

Let us consider for instance the III quadrant, where both $\tilde{s}$ and $\tilde{R}$ are small. The fact that $\tilde{s}$ is small means that from the point of view of the crowd the cylinder is perceived as nearly immobile, which explains the rotational invariant shape of the solution. For small values of $\gamma$, the distance at which an immobile perturbation is felt by the crowd is, as mentioned above, given by the healing length $\xi$. As $\gamma$ increases this length should be compared to $d_{c_s} = c_s/\gamma$, the length scale related to the finitude of the anticipation horizon. Hence, from figure 3 we see that the scale of the density perturbation around the obstacle is given by the smallest between the $d_{c_s}$ and $\xi$. In this case, a large $\gamma$ does not, however, modify the qualitative aspect of the density distribution, which remains rotationally invariant.

Figure 4 then focuses on the I quadrant where both $\tilde{R}$ and $\tilde{s}$ are large. Because $\tilde{s}$ is large, the agents feel that the cylinder moves significantly more rapidly than the speed they can themselves maintain comfortably within the crowd. For low $\gamma$, they would therefore tend to anticipate the obstacle arrival by moving sideways quite early, which explains the low density corridor extending rather far in front of the cylinder in that case, typically at a distance of order $l_s = s\xi/c_s$. The density profile is in this case essentially symmetric, as for $\gamma = 0$ the system is invariant under the symmetry $(t \mapsto -t, y \mapsto -y)$. When $\gamma$ increases, $l_s$ should be compared with $d_s = s/\gamma$, which measure how far from the cylinder agents can foresee its motion. As illustrated on Figure 4 , we see that when $d_s < l_s$, the size of the perturbation in front of the

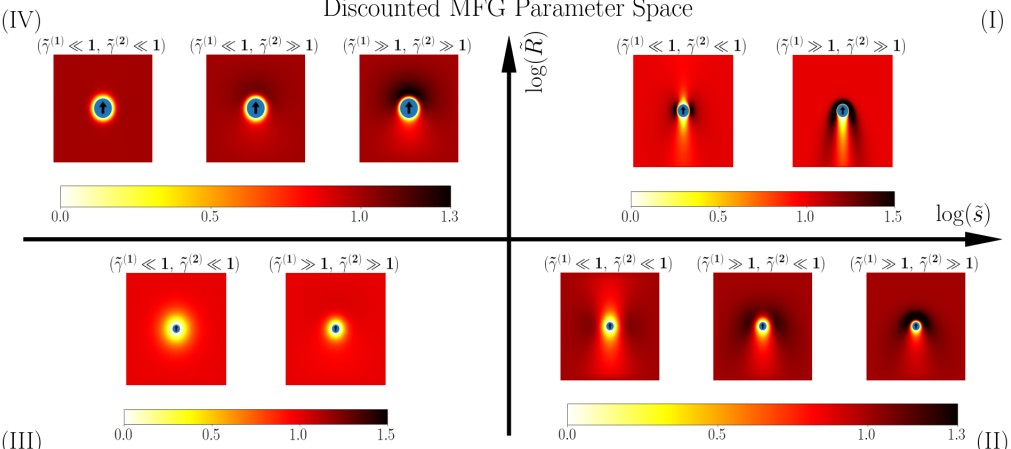

Figure 2: Exploration of parameter space of the MFG model. Each sub-figure represents the discounted stationary MFG density obtained solving (18) and (17). The axis represent the dimensionless quantities $\tilde{R} \equiv R/\xi$ and $\tilde{s} = s/c_s$. Each quadrant shows different values of $\tilde{\gamma}^{(1)} \equiv (\xi/c_s)\gamma$ (or $\tilde{\gamma}^{(2)} \equiv (R/s)\gamma$). We note that the choice of parameters best representing the experimental conditions lies in quadrant I, except for the case of Fig. 1c which lies at the boundary between quadrants I and II. Parameters of the sub-figures in each quadrant in format $\left(\tilde{s}, \tilde{R}; [\tilde{\gamma}^{(1)}]\right)$ [with $\tilde{\gamma}^{(2)} = (\tilde{R}/\tilde{s})\tilde{\gamma}^{(1)}$]: quadrant I $(3, 3; [.25, 5])$, quadrant II $(3, 0.3; [0.5, 5, 40])$, quadrant III $(0.3, 0.3; [0.25, 5])$, quadrant IV $(0.3, 3; [0, 0.45, 1.8])$.

cylinder is given by $l_s$. Contrarily to what happens in the third quadrant however, this change of scale qualitatively alters the solution, with a density blob forming in front of the cylinder and on the other hand a density profile behind the cylinder which is much less affected.

The variations of the density plots seen on quadrant II and IV are somewhat more complex since the four length scales $(\xi, l_s, d_{c_s}, d_s)$ are involved, but the mechanisms observed in Figure 3 and Figure 4 can be seen to be at work there too.

## 4 Comparisons with experiment

We now turn to the analysis of the experimental settings previously discussed. We will deal with the three different experimental configurations showed in Figure 1 separately. Our claim is that we can describe the three scenarios as the stationary state of our MFG model. This is already true for the frontal configuration, as was shown in [1]. Our goal is therefore to demonstrate that this is also true for the other two configurations. We will compare the experimental data to the density field $\hat{m}_e$ and to the velocity field $v(\vec{x}) = -[\mu^{-1}\nabla \hat{u}^e + \sigma^2(\nabla \hat{m}^e)/(2\hat{m}^e)]$ derived from the system Eqs. (18)-(17). For completeness, we have also solved the time dependent MFG equations (HJB)-(KFP) and in appendix C we compare their solution at intermediate times $t \simeq T/2$ to $\hat{m}_e$, and verify that they are as expected identical, the ergodic approach being however significantly faster and more precise.

### 4.1 Pedestrians facing the obstacle

In [1] we studied the case where pedestrians were facing the arriving cylinder. Under these circumstances, people could efficiently estimate the cylinder's size and velocity, thus the time it would take for the obstacle to reach them. We model such scenario by taking $\gamma = 0$, and the

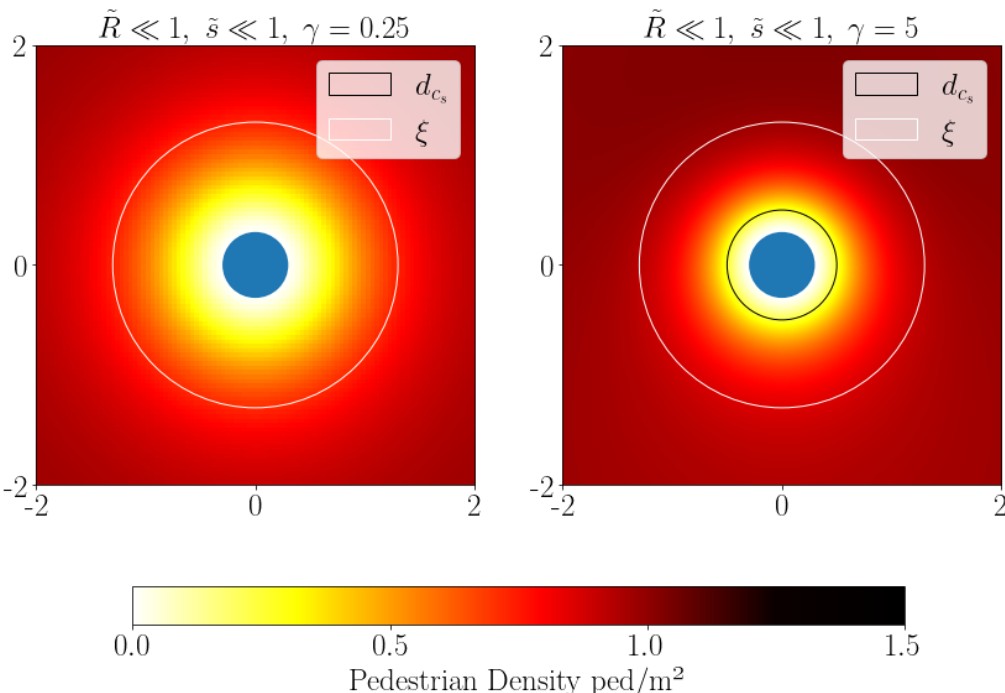

Figure 3: Focus on the quadrant III of Figure 2 with details about the size of the perturbation. The white circle has radius $R + \xi$ whereas the black circle has radius $R + d_{c_s}$, with $d_{c_s} \equiv c_s/\gamma$ (note that the left panel's black circle is not visible since $d_{c_s} = 4$). We observe how the smallest of the circles is the one governing the distance at which the perturbation due to the cylinder is felt.

simulation is shown on the left column of Figure 5. Here we see how striking the resemblance to the experiment is. Starting from the density plot, we clearly see the vertically symmetric distribution of pedestrians, with a depletion prior and posterior to the obstacle passage and with an increase on the sides. Moreover, the velocity field obtained from the MFG model correctly displays the lateral motion of pedestrians stepping aside to make room for the intruder. We believe MFG good performances should be attributed to an element that, as already discussed in [1], many pedestrian dynamics models, such as granular and social force models, struggle accounting for. This is the long term anticipatory behavior of individuals clearly captured in the experiment and naturally present in the very structure of MFG, namely by means of the backward-solved HJB equation (HJB). The back-propagation of information allows MFG agents to optimally anticipate the obstacle's arrival, and most notably *without prescribing an anticipation time*, contrarily to many models for crowds motion. Indeed, solving the MFG equations gives the Nash equilibrium velocity an agent should adopt for any given point in the simulated space, i.e. the strategy of motion which, in the spirit of (2), is best suited to avoid the obstacle and high density areas.

## 4.2 Pedestrians randomly oriented

In the experimental configuration where participants were asked to orient themselves randomly we see, from the right column of Figure 6, that the main difference with the frontal case is in the decrease of the depletion in front of the obstacle, meaning that participants anticipate less. We can imagine that, when participants were placed randomly, only some of them could gather information about the obstacle visually, whereas the rest had to resort to all their other senses to decide how to react. This impacts the global anticipatory behavior and

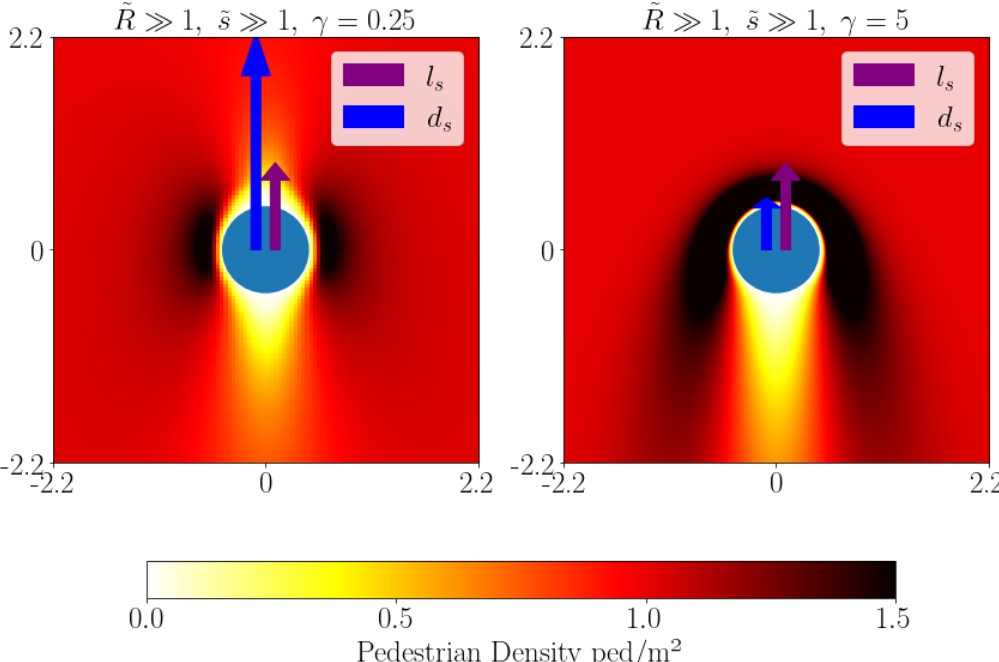

Figure 4: Focus on the quadrant I of Figure 2 with details about the size of the perturbation. The purple arrow's length is $R + l_s$, with $l_s \equiv s\xi/c_s$ the scale of the length at which agents should start moving to optimally avoid impact, whereas the blue arrow's length is $R + d_s$, where $d_s = s/\gamma$ represents how far the cylinder travels during time $1/\gamma$. It is apparent how a congestion in front of the cylinder appears when $d_s < l_s$, because the agents optimize only on a small portion of cylinder's trajectory.

causes a later reaction to the obstacle arrival. The inclusion of the discount factor $\gamma$ in our MFG is enough to describe the change in the crowd's avoidance strategy. The left column of Figure 6 shows the numerical solution of the MFG system for $\xi$ and $c_s$ as for the frontal case and with $\gamma = 0.5$. We can indeed observe that turning on the discount factor produces the desired effects, by reducing the crowd's displacement in front of the obstacle but still conserving the accumulation on the sides and the density depletion after its passage. Moreover, the simulated velocity field shows an increase in escaping dynamics in front of the cylinder and slight circulation around it.

### 4.3 Pedestrians giving their back to the obstacle

Finally, when participants in the experiment had to give their back to the obstacle and were asked not to anticipate, the observed behavior changed decisively. As the right panel of Figure 7 shows, having lost the visual information, it was harder to estimate the obstacle's velocity and direction of motion, resulting in pedestrians being pushed along by the intruder, and behaving like granular material [1]. Behind the cylinder, on the other hand, no significant depletion is shown, and this we believe is due to the diffusivity of the crowd, given the pedestrians' intention to have as much space as possible. The left column of Figure 7 shows the MFG simulation for $\xi = 0.4$, $c_s = 0.2$ and $\gamma = 6$. Here we recover the accumulation in front of the obstacle and the smaller depletion behind it. By looking at the velocity plots we clearly see that pedestrians in front of the cylinder are pushed by the intruder along the direction of its motion. Then, we also remark the agreement with the rotational motion around the obstacle, analogously to what would happen for granular inert matter, under purely mechanical forces. We managed to recover the experimental behavior mainly by means of the discount factor, slightly modifying

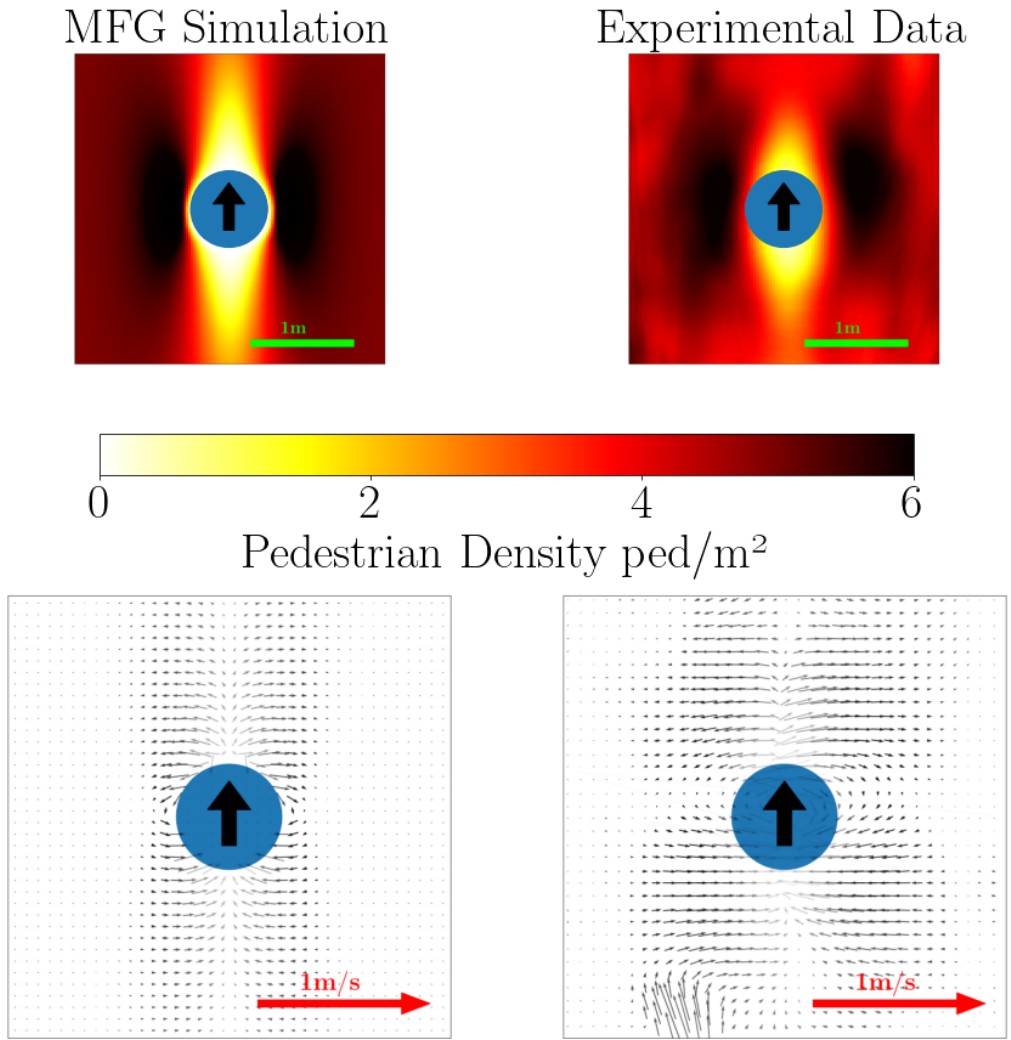

Figure 5: Qualitative comparison between density, first row, and velocity plots, second row, between the experiment, right column, and the ergodic state of the MFG model, with parameters $\xi = 0.2$, $c_s = 0.1$ and $\gamma = 0$, left column, for the case where all pedestrians were facing the incoming intruder. The green segment and the red arrow are a scale for distance and velocity respectively.

$\xi$ and $c_s$ to better fit the data, placing the solution at the boundary between quadrants I and II. This means that the discounting term correctly reproduces losses in anticipatory abilities.

## 5   Conclusion

In the present work we have provided a detailed description of the consequences of adding to the Mean-Field Games model used in [1] a term representing agents anticipation's horizon. Moreover, we have shown that such term helps modulating the simulated anticipatory behavior, making it possible to reproduce a wider range of crowd dynamic scenarios, as the comparison with the different experimental configurations has shown. This is all the more remarkable since we used the most simple and first ever proposed Mean-Field Game model [8], to which we just added a parameter, the discount factor. By using our MFG to describe, both quantitatively and qualitatively, the human behavior observed in this particular experiment,

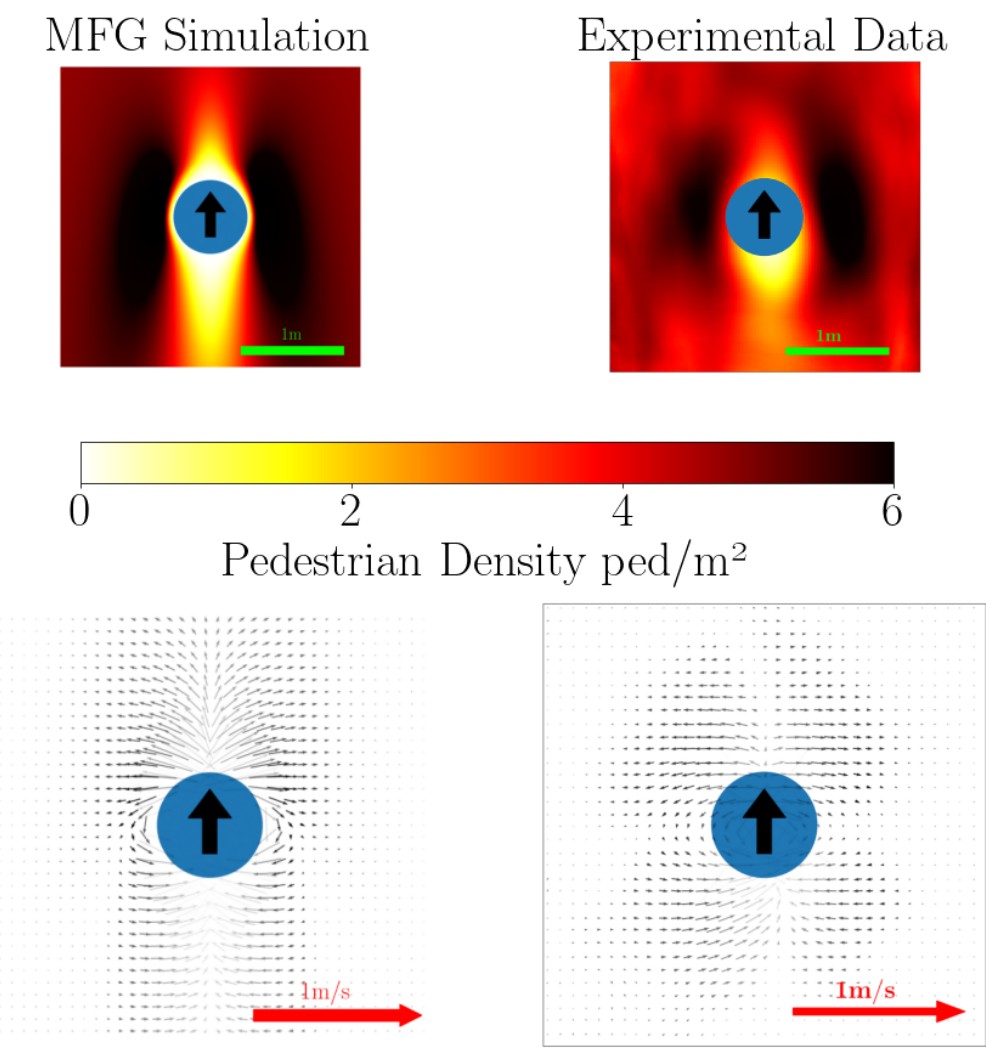

Figure 6: Qualitative comparison between density, first row, and velocity plots, second row, between the experiment, right column, and the ergodic state of the MFG model, with parameters $\xi = 0.2$, $c_s = 0.1$ and $\gamma = 0.5$, left column, for the case where pedestrians were oriented randomly.

we do not wish to imply that pedestrians solve a system of coupled differential equations while walking, but we highlight that at least in simple cases, individuals have internalized some basic anticipation mechanisms allowing them to rapidly coordinate with others and efficiently avoid eventual obstacles.

## Acknowledgements

We acknowledge the precious help of Alexandre Nicolas who prepared and provided the experimental data with the conventions used in this Paper. We thank Alberto Rosso for the discussions about this work and useful comments about the manuscript.

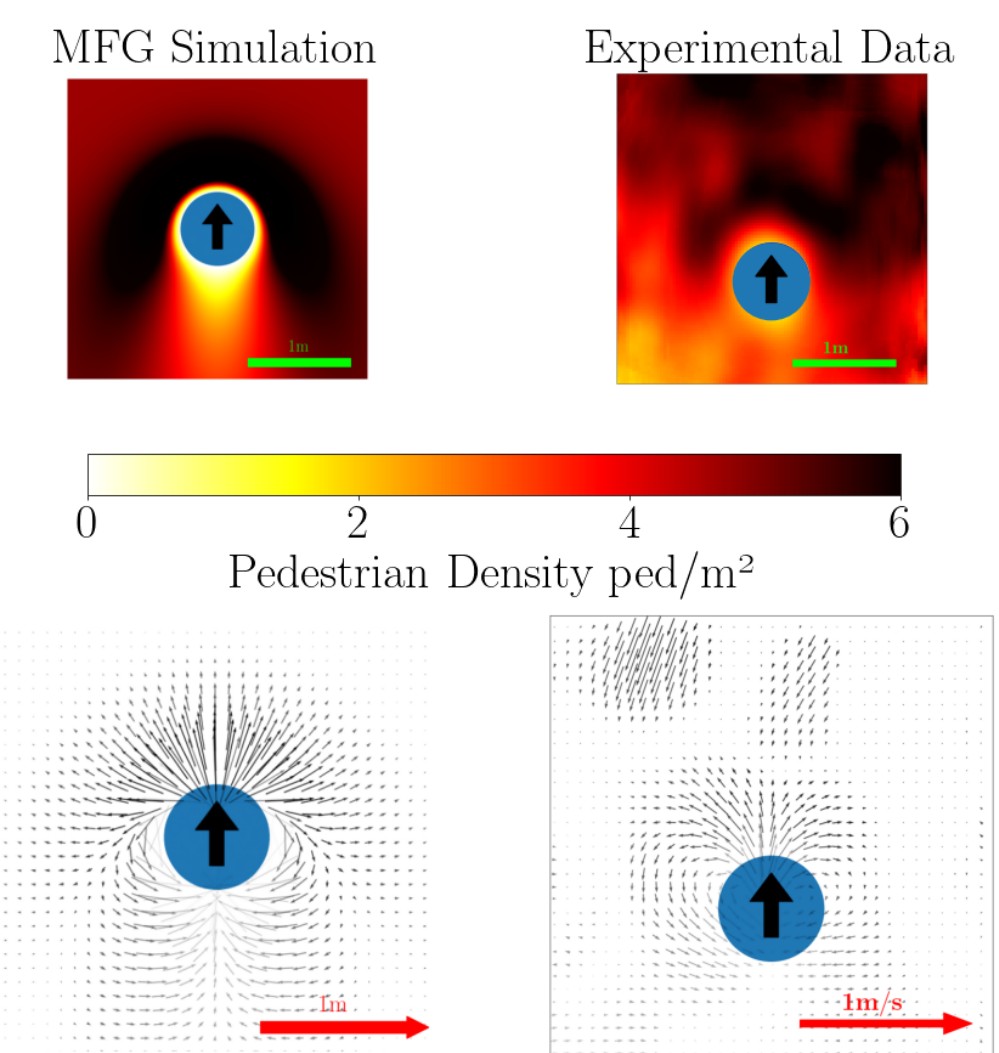

Figure 7: Qualitative comparison between density, first row, and velocity plots, second row, between the experiment, right column, and the ergodic state of the MFG model, with parameters $\xi = 0.4$, $c_s = 0.2$ and $\gamma = 6$, left column, for the case where pedestrians were giving their back to the incoming cylinder and were asked not to anticipate.

# A  Derivation of the discounted HJB

Starting from the definition of the cost functional (2) we have

$$
u(\vec{x}, t) = \inf_{\vec{a}} \mathbb{E} \left\{ \int_t^T \mathcal{L}(\vec{x}, \tau)[m] e^{\gamma(t-\tau)} d\tau + e^{\gamma(t-T)} c_T(\vec{x}_T) \right\}
$$

$$
= \inf_{\vec{a}} \mathbb{E} \left\{ \int_t^{t+dt} \mathcal{L}(\vec{x}, \tau)[m] e^{\gamma(t-\tau)} d\tau + e^{-\gamma dt} \right.
$$

$$
\left. \times \left( \int_{t+dt}^T \mathcal{L}(\vec{x}, \tau)[m] e^{\gamma(t+dt-\tau)} d\tau + e^{\gamma(t+dt-T)} c_T(\vec{x}_T) \right) \right\} .
$$

Making use of the *dynamic programming principle* [19], this can be written as

$$
u(\vec{x}, t) = \inf_{\vec{a}} \left[ \mathcal{L}(\vec{x}, t)[m] dt + e^{-\gamma dt} u(\vec{x} + d\vec{x}, t + dt) \right]
$$

$$
\overset{\text{Ito}}{=} \inf_{\vec{a}} \left\{ \mathcal{L}(\vec{x}, t)[m] dt + (1 - \gamma dt) \left[ u(\vec{x}, t) + dt \left( \partial_t u + \vec{a} \cdot \vec{\nabla} u + \frac{\sigma^2}{2} \Delta u \right) \right] \right\} ,
$$

where in the last passage the Ito chain rule has been used to calculate the total time derivative of the value function. Then, by keeping only terms of order one in $dt$ we obtain

$$
0 = \partial_t u - V[m] + \frac{\sigma^2}{2} \Delta u - \gamma u + \inf_{\vec{a}} \left\{ \frac{\mu}{2} \vec{a}^2 + \vec{a} \cdot \vec{\nabla} u \right\} . \tag{A.1}
$$

At this point, by minimizing the term in the curly brackets with respect to $\vec{a}$ we find that the optimal velocity is given by

$$
\vec{a}^* = -\frac{\vec{\nabla} u}{\mu} . \tag{A.2}
$$

Finally, (A.2) is plugged back in (A.1) to obtain (HJB).

# B  Alternative formulation of MFG

In [1], we took advantage of the fact that by applying the Cole-Hopf transformation

$$
u(\vec{x}, t) = -\mu \sigma^2 \log \Phi(\vec{x}, t), \tag{B.1}
$$

one can cast the MFG equation in a form more familiar to physicists. Now, substituting (B.1) into equation (HJB) we obtain

$$
\mu \sigma^2 \partial_t \Phi = -\frac{\mu \sigma^4}{2} \Delta \Phi - V[m] \Phi + \gamma \mu \sigma^2 \Phi \log \Phi . \tag{B.2}
$$

The second part of this transformation amounts to defining $\Gamma(\vec{x}, t) = \frac{m(\vec{x}, t)}{\Phi(x, t)}$ which, when plugged inside the KFP equation (KFP) yields

$$
\mu \sigma^2 \partial_t \Gamma = \frac{\mu \sigma^4}{2} \Delta \Gamma + V[m] \Gamma - \gamma \mu \sigma^2 \Gamma \log m + \gamma \mu \sigma^2 \Gamma \log \Gamma . \tag{B.3}
$$

The stationary equations are

$$
-\frac{\mu \sigma^4}{2} \Delta \Phi^e - V[m^e] \Phi^e + \gamma \mu \sigma^2 \Phi^e \log \Phi^e = 0, \tag{B.4}
$$

$$
\frac{\mu \sigma^4}{2} \Delta \Gamma^e + V[m^e] \Gamma^e - \gamma \mu \sigma^2 \Gamma^e \log \Phi^e \Gamma^e + \gamma \mu \sigma^2 \Gamma^e \log \Gamma^e = 0. \tag{B.5}
$$

When passing to the moving frame, as in 2.3, the time dependent MFG in the Cole-Hopf settings becomes

$$\mu\sigma^2\partial_t\hat{\Phi} - \mu\sigma^2\vec{s}\cdot\vec{\nabla}\hat{\Phi} = -\frac{\mu\sigma^4}{2}\Delta\hat{\Phi} - V[m]\hat{\Phi} + \gamma\mu\sigma^2\hat{\Phi}\log\hat{\Phi}\,, \tag{B.6}$$

$$\mu\sigma^2\partial_t\hat{\Gamma} - \mu\sigma^2\vec{s}\cdot\vec{\nabla}\hat{\Gamma} = \frac{\mu\sigma^4}{2}\Delta\hat{\Gamma} + V[m]\hat{\Gamma} - \gamma\mu\sigma^2\hat{\Gamma}\log m + \gamma\mu\sigma^2\hat{\Gamma}\log\hat{\Gamma}\,, \tag{B.7}$$

whereas from (B.4) and (B.5) we get the corresponding equations for the stationary state

$$-\frac{\mu\sigma^4}{2}\Delta\hat{\Phi}^e - V[m_0]\hat{\Phi}^e + \gamma\mu\sigma^2\hat{\Phi}^e\log\hat{\Phi}^e + \mu\sigma^2\vec{s}\cdot\vec{\nabla}\hat{\Phi}^e = 0\,, \tag{B.8}$$

$$\frac{\mu\sigma^4}{2}\Delta\hat{\Gamma}^e + V[m_0]\hat{\Gamma}^e - \gamma\mu\sigma^2\hat{\Gamma}^e\log m + \gamma\mu\sigma^2\hat{\Gamma}^e\log\hat{\Gamma}^e + \mu\sigma^2\vec{s}\cdot\vec{\nabla}\hat{\Gamma}^e = 0\,. \tag{B.9}$$

Now, if $\gamma = 0$, we see that these equations are similar to the Non-linear Schrödinger equation in a moving frame, and in [1] we explain that casting the problem in this form is beneficial to the development of efficient numerical schemes, especially for the stationary state of MFG. Unfortunately the presence of the logarithmic term does pose significant problems for the numerical implementation when $\gamma > 0$, if one tries to solve for the stationary state in this formulation. However, as explained in appendix C, this form of the MFG equations is instrumental in curing the divergences occurring near the obstacle in the original $(m, u)$ formulation of the MFG equations.

## C   Brief description of the numerical scheme

Figures 5, 6 and 7 compare the stationary state of our MFG model to the experimental data collected in [7]. The stationary states shown there, as well as in Fig. 2, are obtained solving directly equations (17) and (18), using an algorithm which at its core is related to the one used in [1] for the stationary state. This consists in choosing a grid representing the space, in this case the 2d area where the experiment took place. Then, for each site $i$, both $\hat{m}_i^e$ and $\hat{u}_i^e$ are expressed as functions of neighboring sites. By starting from an initial guess, the values of $\hat{u}^e$ and $\hat{m}^e$ are updated until convergence.

Within the $(\hat{u}^e, \hat{m}^e)$ variables, this approach fails however near hard walls, or here near the cylinder, where the value function $u$ shows a logarithmic divergence. This is why in [1] we applied this method to the equations in their NLS form (i.e. with the variables $(\hat{\Phi}^e, \hat{\Gamma}^e)$). Unfortunately, as explained in appendix B, when $\gamma \neq 0$ the MFG equations in the $(\hat{\Phi}^e, \hat{\Gamma}^e)$ variables imply a logarithm, and this logarithm makes this approach unfeasible.

For finite $\gamma$, it is therefore necessary to mix the solution of the equations in both formulations, and more specifically to solve the problem expressed in the $(\hat{\Phi}^e, \hat{\Gamma}^e)$ variables near the obstacles, and in its original $(\hat{u}^e, \hat{m}^e)$ form away from it. This has proven to be a very reliable and fast algorithm, allowing us to directly compute the solution of the ergodic state. More details on this method will be provided in a future publication.

As a check of coherence, we also have solved the time dependent MFG system, to verify that the solution we obtain solving (18) is the same to what we get by taking the solution of (16) at $t = T/2$. We solved the time dependent problem using the Matlab routine ode45. Figures 8 and 9 show the results of this comparison for the choice of parameters used in Figs. 6 and 7 demonstrating that the two approaches indeed give the same result. The ergodic approach is, however, significantly faster and more precise, in particular because the time-dependent approach, being performed in the lab reference frame, requires a large system.

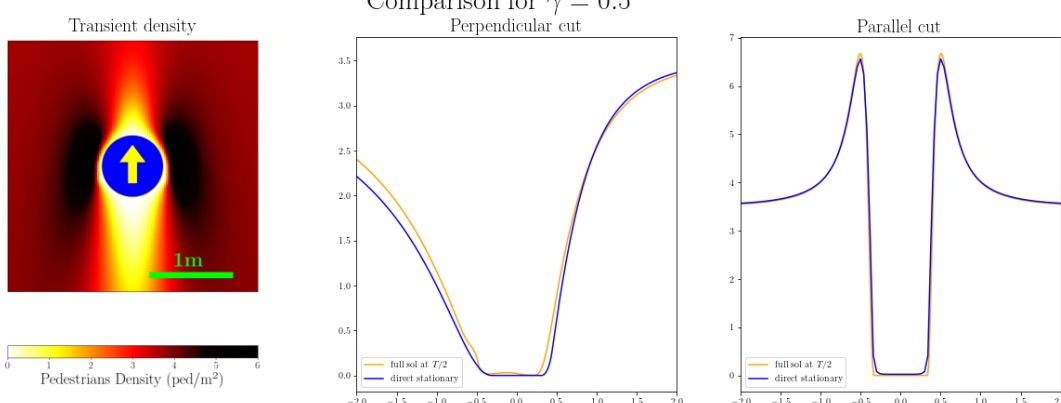

Figure 8: Middle and right panels: density profiles of the discounted MFG stationary state along a vertical (middle) and horizontal (right) cut passing through the center of the cylinder. The curves compare the solution obtained solving directly Eqs. (18)-(17) (blue line) and the one obtained from the time dependent Eqs. (HJB)-(KFP) and taking the solution at $T/2$ (orange line). Left panel: density plot obtained from the time dependent Eqs. (HJB)-(KFP). The parameters of the simulations are the same as in Fig. 6 (corresponding to $\gamma = 0.5$).

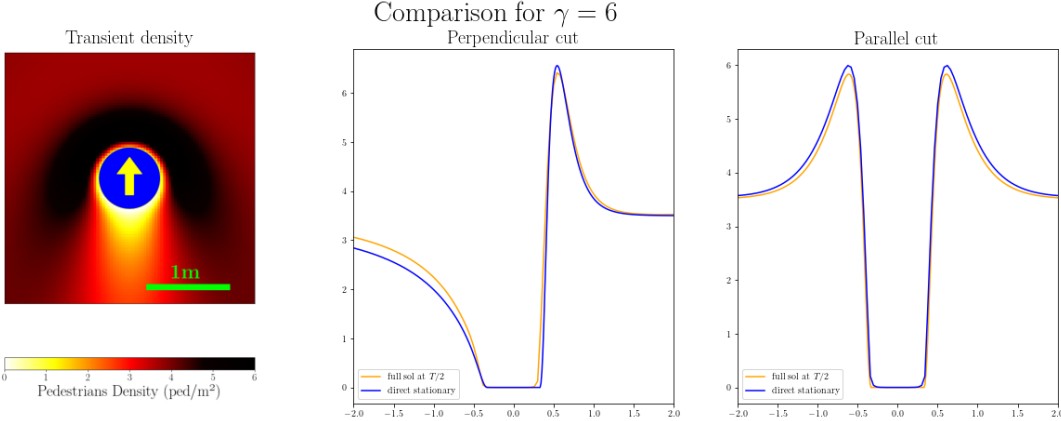

Figure 9: Same as in Fig. 8, but for the parameters of Fig. 7(corresponding to $\gamma = 6$).

## D   Dimensionless equations

In section 3 we have introduced the length scale $\xi$ and velocity scale $c_s$, which are such that at $\gamma = 0$, the solution of the MFG equations are entirely determined by the dimensionless ratio $\tilde{R} = R/\xi$, and $\tilde{s} = s/c_s$.

Considering now $\gamma \neq 0$, let us introduce the scaled variables $\tilde{x} = x/\xi$, and $\tilde{t} = t/\tau$, where $\tau$ is the characteristic time scale of the system obtained as $\tau = \xi/c_s$. Substituting these in (15) we get

$$\frac{\partial_{\tilde{t}} u}{\tau} = -\frac{\sigma^2}{2\xi^2}\Delta_{\tilde{x}\tilde{x}}u + \frac{(\vec{\nabla}_{\tilde{x}}u)^2}{2\mu\xi^2} + \gamma u + \frac{c_s}{\xi}\vec{\tilde{s}} \cdot \vec{\nabla}_{\tilde{x}}u + gm + U_0(\tilde{x}\xi, \tilde{R}\xi). \tag{D.1}$$

Introducing now $\tilde{a}^* = a^*/c_s$ the scaled optimal velocity, we see that if we want to define it as $\tilde{a}^* = -\partial_{\tilde{x}}\tilde{u}$ this implies to define the scaled value function as $\tilde{u} = u/(\mu\xi c_s) = 2u/(\mu\sigma^2)$. Noting furthermore than the definition Eq. (13) of $U_0$ implies $U_0(\tilde{x}\xi, \tilde{R}\xi) = U_0(\tilde{x}, \tilde{R})$, Eq. (D.1)

reads

$$\partial_{\tilde{t}}\tilde{u} = -\Delta_{\tilde{x}\tilde{x}}^2\tilde{u} + \frac{(\vec{\Delta}_{\tilde{x}}\tilde{u})^2}{2} + \tilde{\gamma}^{(1)}\tilde{u} + \vec{\tilde{s}} \cdot \vec{\nabla}_{\tilde{x}}\tilde{u} + \frac{m}{m_0} + U_0(\tilde{x}, \tilde{R}), \tag{D.2}$$

with $\tilde{\gamma}^{(1)} = \gamma\tau$ is the scaled discount ratio. Applying the same scaling to Eq. (16) we obtain the dimensionless KFP equation

$$\partial_{\tilde{t}}m = \Delta_{\tilde{x}\tilde{x}}m + \vec{\nabla}_{\tilde{x}} \cdot (m\vec{\nabla}_{\tilde{x}}\tilde{u}) + \vec{\tilde{s}} \cdot \vec{\nabla}_{\tilde{x}}m. \tag{D.3}$$

Up to a scaling, the solution of our model is therefore entirely determined by the scaled quantities $(\tilde{R}, \tilde{s}, \tilde{\gamma}^{(1)})$, or equivalently $(\tilde{R}, \tilde{s}, \tilde{\gamma}^{(2)})$ where $\tilde{\gamma}^{(2)} = (\tilde{R}/\tilde{s})\tilde{\gamma}^{(1)}$.

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
