# Peer review of "Discounted Mean-Field Game model of a dense static crowd with variable information crossed by an intruder"

_SciPost Physics, doi:SciPost Phys. 16, 104 (2024)_

## Round 1 · Referee Report · Anonymous · 2023-8-22

Strengths
this paper enhances results obtained in
[7] Bonnemain, Thibault, Matteo Butano, Théophile Bonnet, Iñaki Echeverría-Huarte, Antoine Seguin, Alexandre Nicolas, Cécile Appert-Rolland, and Denis Ullmo. "Pedestrians in static crowds are not grains, but game players." Physical Review E 107, no. 2 (2023): 024612.
[6] Nicolas, Alexandre, Marcelo Kuperman, Santiago Ibañez, Sebastián Bouzat, and Cécile Appert-Rolland. "Mechanical response of dense pedestrian crowds to the crossing of intruders." Scientific reports 9, no. 1 (2019): 105.
The first one advocated for a game theoretical modelling approach for crowds, with a specific focus on Mean Field Games (MFG), while the second one focused on experimental results for pedestrian crowds avoiding intruders (or obstacle).
The submitted paper uses MFG to model crowds avoiding intruders, having an "anticipation parameter" corresponding to the discount factor of the stochastic control model for each agent in the crowd. The goal of the paper is to demonstrates that using such a parameter matches the empirical results of paper [6].
Weaknesses
Section 2 is deriving approximated results in two steps
- ergodic version of the problem with a discount factor $\gamma$
- introducing a moving obstacle in the derived ergodic version of the problem.
Nevertheless in section 3 (ie for comparison of numerical simulations with experiments), the formula of section 2 are never used.
If nothing changes in the paper, section 2 is not really useful, and I am not sure that without any derived approximation, the comparison deserves the publication of a paper. At least the structure of the paper should change.
They is a list of small remarks like: adequacy of the citations, assumptions that are made, caption of the figure, etc. That deserve to be corrected. (see later).
Report
The main weakness of the paper today is that derived results are not used for numerical comparisons. Since different steps of the results rely on assumptions, nothing in the paper supports these assumptions any more (if the approximations would match the observation, or even the numerical simulation, it would demonstrate that they are adequate).
Because of that I do not recommend the publication of the submitted paper as it is. I consider my Suggestions A and B to be a major revision request; it is nevertheless reasonable to think authors can match these expectations.
Suggestion A. Nevertheless I think some changes can be done to add value to it; the main modification could be to focus on the emergence of an ergodic regime.
In the paper, author claims (Section 2.2) that "for times sufficiently larger than the initial time and smaller than $T$ it has been shown [...] that for a generic MFG model a stationary or ergodic state exists." Then in Appendix C they explain they did not used their approximations of the simulation and add "we can therefore access the ergodic state of the system for $t = T /2$" since they are interested in this ergodic regime.
I would suggest that authors look at the value functions and the mean field in their simulations "around $T/2$" (for instance for all $kT/K$ for $K=10$ and $k\in\{3,4,5,6,7\}$ and show numerically that they are very close around $k=5$, to demonstrate that an ergodic state exists for this problem (they may have to take $K\simeq 20$ and adjust $k$ that for).
Suggestion B. Ideally, since author access to simulations of the exact problem, they could check that some of their approximations are correct. For instance looking at $\partial_t u$ and comparing it with their $\lambda$ (in one of their approximation $\partial_t u\propto \lambda$), their choice of approximation of the function $f(t)= ke^{\gamma t} + \lambda/\gamma$ can be checked too around the ergodic regime that I suggest to identify numerically in Suggestion A.
Requested changes
In my opinion it is very difficult to maintain the paper as it is without answering to Suggestion A.
Suggestion B would add value to Section 2 of the paper, otherwise it is difficult to know if the assumptions taken by the authors are valid.
Other adjustments:
Adjustment 1. In Section 2.2, author write "it has been shown, for example in [8], that for a generic MFG model a stationary or ergodic state exists [...] so that we can write $u(x, t) ≃ u_e(x) + f (t)\quad (10)$."
I do not find any such decomposition in the reference [8] J.-M. Lasry and P.-L. Lions, Jeux à champ moyen. i – le cas stationnaire, C. R. Acad. Sci. Paris 343(9), 619 (2006).
=> Authors should change the reference such that it matches their formula (10).
Adjustment 2. Author write "Our hypothesis is that, in the case of $\gamma > 0$, (10) is still valid, thus, by plugging it in (8), we obtain...". Authors may check if it is the case in their numerical simulations (that is compatible with Suggestion B).
Typo-like comments:
- Explaining that for formula (4) $g m(x,t)+U_0(x,t)$, the first term corresponds to avoidance of concentration by the population and the second term corresponds to the impossibility to be inside the cylinder would help the readability of the paper. As of today the reader has to wait for eq (15) to understand the role of each term.
- At least the caption of one of the Figures should explain that the red arrow is a velocity scale (cf. Figure 4 of [6] ). It is not mentioned anywhere in the current version of the paper.

---

## Round 1 · Referee Report · Anonymous · 2023-9-11

Strengths
1. Clear and well written
2. Convincing results
Weaknesses
1. The structure of the manuscript is unsuited
2. Theoretical results unused
3. Simplifying assumptions unjustified and untested
4. The fraction of ne results in the manuscript is low
Report
This manuscript focuses on the dynamics of pedestrians avoiding a moving obstacle. It is structured in two distinct parts: The extension of the mean-field game theory publihed in [7] with a "discount" factor, which accounts for the anticipation of agents, and the qualitative comparison of this model with experiments published in [6].
This manuscript is written in a clear and concise manner, proposes a novel and promising approach to the modeling of pedestrian dynamics, and the density and velocity fields comparison between the presented model and the experiments is convincing.
The current structure of the manuscript, however, is puzzling. Part 2 formulates an extension of the MFG theory developped in [7], Eqs. (2) to (9), simplifies it in the asumption that the density is time-independent, Eqs. (10) to (14), and then transposes it to a moving frame, Eqs.(15) to (20). Part 3, surprisingly, numerically solves the full model Eqs.(2) to (9), and does not use any of the simplifications and assumptions developped in Part 2.
This has two major consequences for the manucsript.
(1) The reader is left wondering why the simplifications in Part 2 were presented in the manuscript in the first place.
(2) More importantly the major simplifying asumption in Part 2, the time-independence of the density field, is neither a priori justified theoretically nor a posteriori validated numerically. As a result, it is impossible to assess the relevance of Parts 2.2 and 2.3
That being said, all is left in the manuscript is the initial description of this new MFG theory model and the quantitative comparison with experimental data. In my humble opinion, this seems too little to be published as a scientific paper as is.
There are, however, some major changes that could make the manuscript much stronger:
- following my previous comments, the structure of the manucsript should be fully revised
- the authors could explore how the addition of the discount factor modifies or not the four quadrant repartition discussed in ref [7]
- the manuscript could show whether the theoretical limit of \gamma -> Inf converges to a model that is compatible with the pedestrians showing their back to the moving obstacle
- the ergodicity assumption could be assessed numerically, and the manuscript clearly show if the density field time-independent in this model
To conclude, this manuscript is very well written and presents worthy results, but I think it needs to be significatly revised before publication.
Requested changes
1. following my previous comments, the structure of the manucsript should be fully revised
2. the authors could explore how the addition of the discount factor modifies or not the four quadrant repartition discussed in ref [7]
3. the manuscript could show whether the theoretical limit of \gamma -> Inf converges to a model that is compatible with the pedestrians showing their back to the moving obstacle
4. the ergodicity assumption could be assessed numerically, and the manuscript clearly show if the density field time-independent in this model
Additional minor remarks:
5. There is a typo on page 6: small values of the discount factor should be \gamma -> 0, not \gamma -> Inf.
6. In my opinion, the sentence "We strongly believe in the validity of our approach" and following does not belong in a physics paper. A manuscript formulates and tests hypotheses, it does not claim beliefs.

---

## Round 2 · Referee Report · Anonymous (Referee 2) · 2024-1-17

Report
I thank the authors for the efforts they put into revising their manuscript. They took into account my remarks, and responded in great details. I appreciate hte inclusion of the four-quadrants comparison and their efforts in discussing the impact of the discount factor on it, as well as the numerical derivation of the time-independent calculations. Although the latter does not change the conclusions about the comparison to experiments, it justifies the theoretical development presented in Section 2.
Although I am happy to support publication of this manuscript, I have two additional remarks I would like the authors to seriously consider
-
I don't find the "discount term" particularly intuitive, as it is an inverse time. Would it be meaningful to use its inverse, as an anticipatory time scale? One could then argue that the classical MFG theory is for infinite anticipation, and that this work looks into finite anticipation times (such as when pedestrians have their back to the obstacles). I would find such a description simpler than arguing for a cost of actions, more intuitive, and probably capable of reaching a larger community of scientists interested in crowds dynamics
-
I reiterate that, in my opinion, claims such as "We strongly believe in the validity of our approach" are inadequate and do not serve the manuscript well. Instead, saying that "The strong agreement between our model and experimental data supports the idea that, although people do not solve non-linear systems while walking, their basic anticipation mechanisms allows them to, etc" would be both more rigorous and convincing.
Requested changes
See above

Author: Matteo Butano on 2024-02-08 [id 4303]
(in reply to Report 1 on 2024-01-17)We thank the referee for taking the time to read and review the latest version of our manuscript. We welcome his/her appreciation for our modifications to it, and we acknowledge his/her remarks. We have modified the manuscript accordingly to the best of our ability and highlighted such modifications in blue in the attached text. In the following we answer in detail the referee's remarks.
"I don't find the "discount term" particularly intuitive, as it is an inverse time. Would it be meaningful to use its inverse, as an anticipatory time scale? One could then argue that the classical MFG theory is for infinite anticipation, and that this work looks into finite anticipation times (such as when pedestrians have their back to the obstacles). I would find such a description simpler than arguing for a cost of actions, more intuitive, and probably capable of reaching a larger community of scientists interested in crowds dynamics"
"I reiterate that, in my opinion, claims such as "We strongly believe in the validity of our approach" are inadequate and do not serve the manuscript well. Instead, saying that "The strong agreement between our model and experimental data supports the idea that, although people do not solve non-linear systems while walking, their basic anticipation mechanisms allows them to, etc. " would be both more rigorous and convincing. "
Attachment:
colored_manu.pdf

---

## Round 2 · Author Response

Dear SciPost Editor,
We thank you for forwarding the reports of the referees concerning our submission "Discounted Mean-Field Game model of a dense static crowd with variable information crossed by an intruder". We understand that the referees found positive qualities to our work, referee I in particular stating that it was "clear and well written", and has "convincing results". Both, however, considered that it had flaws that had to be corrected before publication. Both referees expressed the concern that we derive time independent equations, (Eqs.\ (10) to (14) which are then transposed in the moving frame as Eqs.\ (15) to (20)), which are not used in practice when we turn to the numerical implementation of the model. This furthermore had two consequences, the first one being that one may wonder why they are introduced, and the second being that the existence of an "ergodic/time-independent" regime is postulated, but in fact never demonstrated.
We admit that the reason we were not including numerics for the time-independent equations was that we did not know how to do this at the time of our first submission. The referees' comments on this point gave us the incentive to investigate that question again, and it turns out that we were able to solve that problem. The approach that we have used is quite non-trivial (we had to use a mixed representation between the (u,m) variable far from the cylinder and the (Phi,Gamma) close to it) and is briefly discussed in an appendix C.
With this tool now at hands, we have significantly modified the content of our paper, and in particular: - All the simulations in the core of the paper are now made in the "ergodic/time-independent" framework. - We have checked in appendix C that the solution found in this way indeed corresponded to what was obtained at t=T/2 for the time dependent calculation, demonstrating in this way the correctness of our hypothesis about the ergodic solution. - As suggested by referee 1, we have added a discussion on how the discount factor modifies the four-quadrant repartition discussed in our previous paper [Bonnemain et al., Phys. Rev. E 107, 024612 (2023)]
We feel that with these new results, as well as the significant rewriting of the manuscript, we have answered all comments and suggestions of both referees, and we hope our paper is now ready for publication in SciPost.

---

## Round 2 · List of Changes

- A brief discussion of the role of $U_0$ has been introduced in the paragraph following Eq. (4).
- Section 2.2 has been completely redrafted and is now split into two subsections (2.2.1 & 2.2.2).
- A new section 3 has been introduced to discuss the parameter space of the model, with three new figures (Figs. 2,3 &4).
- The first paragraph of section 4 has been modified to stress the fact that we now do numerical computations in the time-independent framework.
- Details of the derivation of the discounted HJB equation have been moved from section 2.1 to a new appendix A.
- Old appendix A is now appendix B
- Old appendix B has been suppressed.
- Appendix C, which briefly describes the numerics, has been completely rewritten, and contains now two figures (Figs.8 & 9) comparing the density profiles obtained with our new numerical scheme with the ones obtained from the time-dependent equations.
- Appendix D introduces the scaled version of the MFG equations and identifies the relevant parameter of the model.

---

## Round 3 · Author Response

We have answered the referee's remarks and we have attached to the related comment, as done previously, a copy of the manuscript where the corresponding changes are highlighted in blue.
Best remarks
Matteo Butano, Cécile Appert-Rolland, Denis Ullmo

---

## Editorial Decision

published